# Deep Learning-Based Comparative Prediction and Functional Analysis of Intrinsically Disordered Regions in SARS-CoV-2

**DOI:** 10.3390/ijms26073411

**Published:** 2025-04-05

**Authors:** Sidra Ilyas, Abdul Manan, Donghun Lee

**Affiliations:** 1Department of Herbal Pharmacology, College of Korean Medicine, Gachon University, 1342 Seongnamdae-ro, Sujeong-gu, Seongnam-si 13120, Republic of Korea; 2Department of Molecular Science and Technology, Ajou University, Suwon 16499, Republic of Korea; mananriaz012@gmail.com

**Keywords:** COVID-19, intrinsically disordered regions, deep learning, drug design, proteome, immune modulation

## Abstract

This study explores the role of intrinsically disordered regions (IDRs) in the SARS-CoV-2 proteome and their potential as targets for small-molecule drug discovery. Experimentally validated intrinsic disordered regions from the literature were utilized to assess the prediction of intrinsic disorder across a selection of SARS-CoV-2 proteins. The disorder propensities of proteins using four deep learning-based disorder prediction models: ADOPT, PONDR^®^VLXT, PONDR^®^VSL2, and flDPnn, were analyzed. ADOPT, VSL2, and VLXT identified a flexible linker (129–147), while VSL2 and VLXT predicted disorder in the Cu(II) binding region (163–167) of NSP1. ADOPT did not predict disordered regions in NSP11; however, VSL2 and VLXT identified disorder in the experimentally validated regions. The IDR in ORF3a is crucial for protein localization and immune modulation, affecting inflammatory pathways. VSL2 predicted significant disorder in the N-terminal domain (18–23), which aligns with experimental data (1–41), overlapping with the TRAF-binding motif, while ADOPT indicated high disorder in the C-terminal domain (255–275), consistent with VSL2 and flDPnn. All tools identified disorder in the N-terminal (1–68), central linker (181–248), and C-terminal (370–419) regions of the nucleocapsid (N) protein, suggesting flexibility and accuracy. The S2 subunit of the spike protein displayed more predicted disorder than the S1 subunit across ADOPT, VSL2, and flDPnn. These IDRs are essential for viral functions, like protein localization, immune modulation, receptor binding, and membrane fusion. This study highlights the importance of IDR in modulating key inflammatory pathways, suggesting that they could serve as promising targets for small-molecule drug development to combat COVID-19.

## 1. Introduction

The global impact of SARS-CoV-2, the strain responsible for the COVID-19 pandemic, has underscored the urgent need for effective antiviral therapies. While vaccines and some therapeutic interventions have been developed, the emergence of new variants and the complexity of the viral pathogenesis highlight the ongoing necessity for effective antiviral drugs [1]. Intrinsically disordered regions (IDRs) have been identified as promising drug targets for mitigating viral replication and immune response against the virus [2]. IDRs are protein segments that lack stable 3D structures under physiological conditions, but play crucial roles in mediating protein–protein interactions, regulation, and signal transduction pathways [3,4]. These regions have been implicated in several viral functions, including replication, immune evasion, pathogenicity, and modulation of host cell machinery, making them attractive candidates for therapeutic intervention [5,6,7,8]. By adopting multiple conformations, interacting with diverse partners, the structural flexibility/plasticity of IDPs plays a variety of roles in controlling cellular processes, including replication, signaling, apoptosis, mRNA processing, DNA condensation, formation of aggregates, and transcription regulation (Figure 1) [9].

Molecular recognition sites, molecular assembly, protein modification, and entropic chains are key features of intrinsically disordered regions (IDRs) in proteins. Entropic chains include flexible linkers, which allow the movement of domains positioned on either end of the linker relative to each other, and spacers that regulate the distances between domains; both play crucial roles in facilitating the functional versatility of IDRs. These regions often serve as display sites for post-translational modifications (PTMs), which increase the functional states in which a protein can exist within the cell [10,11]. IDRs tend to evolve more rapidly than structured protein domains, resulting in functional diversity and adaptive potential. Consequently, identifying homologous regions in IDRs is more challenging than that for structured domains, complicating the prediction of their functions and hindering the transfer of functional knowledge between homologs. This complexity is particularly evident when considering the role of IDRs as drug targets in SARS-CoV-2, where disordered regions are crucial for viral replication and immune evasion, and host cell interactions.

For instance, replicase polyprotein 1ab, essential for viral replication, has disordered regions that help to mediate interactions with the host cell machinery and viral proteins, driving the replication process. The spike (S) glycoprotein, which is responsible for binding to the ACE2 receptor on host cells, contains IDRs that facilitate its structural flexibility, enabling viral entry and fusion with the host membranes [12,13]. The ORF3a protein, implicated in immune evasion and modulating host cell functions, also features IDRs that contribute to its functional diversity and interaction with cellular components. Finally, the nucleocapsid (N) protein, which packages viral RNA, contains IDRs that allow it to bind dynamically with RNA and interact with other viral and host factors, ensuring proper viral RNA processing and immune modulation [14,15,16]. These disordered regions in SARS-CoV-2 proteins underscore the virus’s adaptability and its ability to efficiently hijack host cellular processes for replication and immune escape, making them attractive targets for therapeutic intervention.

In this study, we focus on identifying and analyzing the critical role of the dynamic nature of intrinsic disorder regions (IDRs) present in SARS-CoV-2 proteins using several disorder prediction tools, based on experimentally validated data from the literature. We employed four advanced disorder prediction models, ADOPT, PONDR^®^VLXT and PONDR^®^VSL2, and flDPnn, to assess the disorder propensity of each residue in key proteins. These tools were used to map out the IDRs that are most likely to play a role in viral infectivity, immune modulation, and host protein interactions. By systematically analyzing the IDR profiles, regions that could serve as novel targets for antiviral drug development were identified [17]. The high functional relevance of IDRs in viral lifecycle processes, ranging from host receptor binding to immune modulation, makes them especially appealing for therapeutic intervention [18]. Therefore, this study provides insights into how IDRs can be exploited to inform the development of new drugs and therapeutic strategies aimed at curbing the ongoing pandemic and mitigating future outbreaks. These approaches not only provide a foundation for therapeutic development against SARS-CoV-2, but also establish a framework for targeting similar IDRs in other pathogens.

## 2. Results and Discussion

ADOPT, VLXT, VSL2, and flDPnn were employed to predict IDRs in a set of protein sequences with experimentally validated data. Each model utilizes distinct deep-learning-based models to evaluate the disorder propensity of amino acid residues within the sequences. The disorder predictions for key residues across different models and experimental data are presented in detail in Appendix A. A significant positive correlation was observed between predicted disorder scores and experimentally validated disorder regions for all models. The statistical significance of the model predictions was determined using the chi-squared test (Appendix A). The observed variations in performance suggest that the choice of prediction model may depend on the algorithm used and data trained (Appendix A).

### 2.1. Replicase Polyprotein 1ab

The ORF 1ab genes, which make up two-thirds of the SARS-CoV-2 genome, produce two distinct replicase polyproteins, which further break down by proteases to produce one to sixteen non-structural proteins (NSPs), which aid in transcription, replication, assembly, and packing. The replicase polyprotein 1ab, comprising 7076 amino acids, contains several IDRs that are integral to viral replication. Despite only about 4.6% of its sequence lacking a fixed 3D structure, the disordered regions play critical roles in various aspects of the viral life cycle, including translational inhibition, RNA binding, and interactions with the host machinery. An essential constituent of replicase is NSP1, which has unique structural and functional domains. In order to bind with the host ribosome and specifically target the mRNA entry channel, NSP1’s N-terminus is essential. A crucial stage in viral replication, host protein synthesis, is interfered with by this interaction. On the other hand, NSP1’s C-terminus has a helical shape that strengthens its bond with the ribosome and increases host translation inhibition. Although less well-defined structurally linker regions, which connect these functional domains, are probably crucial in regulating the flexibility and dynamics of NSP1, which affects how it interacts with the ribosome and viral RNA. These linker sections might also support the protein’s overall stability and appropriate folding. The ADOPT model identified the flexible linker region of NSP1 (129–147), while VLXT, VSL2, and flDPnn predicted a high degree of disorder in the NSP1 Cu(II) binding region (163–167). The N-terminal domain (1–300) showed higher predicted disorder compared with the C-terminal domain (3705–4405) in the ADOPT model (Figure 2). Interestingly, VLXT and VSL2 consistently predicted higher disorder scores, especially within the linker and C-terminal domains, indicating potential differences in sensitivity and specificity among the prediction tools.

The experimental flexible linker disorder region (129–147) was partially captured by VSL2 (126–140), VLXT (121–128), and ADOPT (126–136), while flDPnn predicted a different set of residues (156, 158, and 163) that did not overlap with experimental findings (Figure 2). Similarly, the experimentally validated disorder segment (163–167) was identified by VLXT and VSL2 and missed by ADOPT and flDPnn, which predicted a nearby set of residues. For the segment 3501–4200, VSL2 and VLXT demonstrated the highest agreement with experimental data, successfully identifying the experimentally validated region 3982–4007, which was missed by the ADOPT and flDPnn. ADOPT provided the most fragmented predictions, detecting only short segments such as 3503 and 4198–4200, with minimal overlap with experimental data. While VSL2, VLXT, and ADOPT showed strong alignment with segment 1–300, VSL2 and VLXT performed better for region 3501–4200.

### 2.2. NSP1 Flexible Linker Region

The NSP1 protein contains a flexible linker/spacer (129–147) that functions as a structural connector between the unstructured N-terminal domain (NTD) and the C-terminal domain (CTD) of the protein [19]. This linker region exhibits significant intrinsic disorder, allowing it to undergo conformational changes that are essential for NSP1’s role in inhibiting host cell translation. Specifically, NSP1 binds to ribosomal mRNA channels, blocking the initiation of translational and thereby inhibiting host protein synthesis to favor viral replication. Given the critical role of NSP1 in suppressing host immune responses and promoting viral protein synthesis, targeting this linker region presents a promising strategy for inhibiting SARS-CoV-2 replication. Targeting flexible linker regions of NSP1 (129–147) by small molecules could inhibit its interaction with the ribosome, disrupting viral proteins, thereby impeding its translation inhibition function. Small molecules designed to stabilize/destabilize specific conformations of this disordered region could effectively modulate NSP1 activity and limit viral replication.

### 2.3. NSP1 Cu(II) Binding Region

Another disordered region within NSP1, located within segment 163–167, is involved in Cu(II) binding. This region contains key residues, such as W161 and H165, which play a critical role in the protein’s fluorescence response to Cu(II) ions [20]. The binding of Cu(II) induces a significant reduction in fluorescence intensity, suggesting that Cu(II) binding alters the structural dynamics of this region, likely affecting NSP1’s overall function and stability. Small molecules can interfere with Cu(II) binding in NSP1 or the RNA binding activity of the polymerase complex. Given the role of this disordered region in Cu(II) binding, it presents a potential target for therapeutic intervention. Small molecules that mimic or disrupt Cu(II) binding could modulate NSP1 activity, impacting its function. In particular, interfering with Cu(II) binding could impair NSP1’s ability to interact with the host cellular machinery, potentially reducing viral replication or altering its virulence. This flexibility and unique binding properties make it an attractive candidate for small-molecule drug development aimed at SARS-CoV-2.

### 2.4. RNA Binding Region in Polymerase

A disordered region in the SARS-CoV-2 polymerase, spanning residues 3982–4007, is integral to the RNA binding function of the viral replicase complex. This region extends from the NSP8 subunit and facilitates the interaction between the polymerase and RNA backbone, which is crucial for RNA replication. The positively charged residues within the region stabilize the interaction with the negatively charged RNA, enabling efficient binding and polymerization [21] By inhibiting the RNA polymerase interaction, small molecules could block viral RNA synthesis, presenting a valuable therapeutic strategy for targeting SARS-CoV-2.

### 2.5. NSP6 Lipid Binding Region

The disordered region of membrane NSP6 (3660–3681) plays a critical role in lipid binding, which is essential for the viral replication process, including the formation of double-membrane vesicles (DMVs) in the host, and facilitates viral genome replication [22]. This region contains tryptophan residues that act as a fluorescent probe to investigate the structural changes in response to environmental factors, such as the presence of SDS and TFE. Fluorescence shifts observed in these experiments indicate that the NSP6 lipid-binding region undergoes conformational changes when interacting with lipid membranes, contributing to the protein’s function in viral assembly and membrane interactions. The lipid-binding region of NSP6 also presents an attractive target for small molecule inhibitors that could block viral assembly and replication by hindering the formation of membrane-bound vesicles. This IDR region provides structural flexibility and involvement in viral–host membrane interactions that make it a key candidate for drug development aimed at blocking viral replication.

### 2.6. Polyprotein 1a

IDRs within replicase polyprotein 1a comprising only 0.27% play a pivotal role in the interaction of this NSP11 with lipid membranes, a critical step in viral replication and assembly. The ADOPT and flDPnn models did not predict any disordered regions, while VSL2 identified the highest degree of disorder in region 4401–4405, which aligned well with the experimentally validated disordered region (4393–4405). VLXT predictions did not align well with the experimental data, suggesting potential differences in sensitivity and specificity in the prediction tools (Figure 3). The disordered region of NSP11 (4393–4405) exhibits a dynamic disorder-to-order transition in response to environmental factors, such as lipid-like conditions, highlighting the adaptability and functional significance. Circular dichroism (CD) spectroscopy revealed that the C-terminal region (4393–4405) of NSP11 exists in a disordered state in an aqueous solution, but undergoes a transition to an ordered α-helical state in the presence of detergent [23] This transition suggests that this region may function as a lipid-binding motif, facilitating the interaction of NSP11 with the host cell membrane. Lipid-binding proteins are central to many aspects of viral life cycles, including membrane fusion, protein–protein interactions, and the assembly of replication machinery on host cell membranes. The α-helical propensity observed in this disordered region aligns with its functional role in viral replication, assembly, and membrane interactions. The structural transition suggests that this region’s conformational flexibility is crucial for interaction with lipid membranes, which in turn may play a role in anchoring NSP11 to the cellular membranes. This interaction is an important step in viral replication and assembly, as membrane-bound viral proteins are key to the formation of replication complexes and viral progeny.

### 2.7. Spike Glycoprotein

The SARS-CoV-2 spike glycoprotein is known for its high degree of intrinsic disorder, with approximately 27.7% of its sequence (1273 amino acids) exhibiting dynamic structural properties. Using the four disorder prediction tools, we observed that all models were able to predict disordered regions across the spike protein, with notable consistency in the identification of key functional sites. These disordered regions play essential roles in the protein’s functional plasticity, receptor-binding, and immune evasion. Distinct patterns of predicted disorder were observed across the S1 (1–685) and S2 (686–1273) subunits. Notably, the S2 subunit exhibited higher predicted disorder compared with the S1 subunit across all models (Figure 4). Within the S1 subunit, regions of interest, such as the signal peptide (1–26) and regulatory regions (437–508, 681–686), showed elevated disorder scores by Pondr, VSL2, and flDPnn models, suggesting potential flexibility and conformational dynamics crucial for receptor binding and viral entry. Similarly, the S2 subunit, which mediates membrane fusion, also exhibited higher predicted disorder by the ADOPT and VSL2 models, consistent with its role in forming stable protein–protein interactions whereas VLXT performed with good accuracy. Interestingly, regions associated with glycosylation display sites (16–21, 73–78, 148–153, 1157–1162, 1172–1177, and 1194–1198) and limited proteolysis display sites (680–686) displayed varying levels of predicted disorder across the different models, indicating potential interplay between disorder and post-translational modifications. The experimental data highlighted several disorder regions in the spike protein, including the signal peptide region (1–26) that facilitates the targeting of the spike protein to the membrane. Its disordered nature contributes to flexibility during protein trafficking, assisting in its efficient membrane insertion and processing.

The comparison between experimentally validated data and the predictions from ADOPT, flDPnn, VLXT, and VSL2 demonstrates variability in disorder prediction accuracy across models. The experimental data identified disordered segments (16–21, 73–78, 148–153, 437–508, and 681–685), providing a benchmark for evaluating the predictive performance of the computational models. ADOPT showed the most extensive disorder predictions, capturing 1–11, 676–685, and partially overlapping with the experimentally validated 681–685 region. However, it failed to identify key disordered segments, such as 437–508. flDPnn did not provide any predictions, whereas VSL2 predicted scattered disordered regions, including 74–78, 149–154, and 184–186, with partial alignment to 73–78 and 148–153, but lacked coverage of longer disordered segments. VLXT performed similarly, predicting 17–20, 468–475, 601–608, 675–685, and 442–446, aligning partially with the experimental disorder data at specific sites.

The experimental data identified disorder-prone regions within the sequence range of 686–1275, providing a reference for evaluating the predictive models. Among the models, ADOPT demonstrated broad disorder predictions, identifying regions such as 686–688, 695, 699–701, 703, and 724–726. While ADOPT captured some experimentally validated disorder-prone segments, such as partial overlap with 686 and 1157–1162, it did not fully align with longer disordered regions, like 1172–1177. VSL2 exhibited more localized disorder predictions, including 687–691, 703–704, 806–815, and 934–950, partially overlapping with experimentally validated disorder segments. However, it also predicted several regions (1023 and 1174–1194) that were not experimentally confirmed, suggesting possible over-predictions. VLXT provided some overlap with experimental findings, particularly in the 697–709 and 869–871 regions, but did not detect key experimentally validated segments, such as 1157–1162. Meanwhile, flDPnn showed limited disorder prediction. Overall, ADOPT, VSL2, and VLXT exhibited the most extensive disorder predictions, aligning more closely with experimental findings.

Receptor binding domain (RBD) (319–541), a crucial region for ACE2 receptor binding, exhibited a disorder-to-order transition upon binding to ACE2 receptor, as confirmed by cryogenic electron microscopy (cryo-EM). In its unbound state, the RBD is flexible and disordered, allowing for conformational changes necessary for receptor interaction. Upon ACE2 binding, it adopts a well-ordered structure, facilitating viral entry into the host cell. This transition from disorder to order is an attractive target for drug development. Small molecules that stabilize the disordered form of the RBD could inhibit ACE2 binding and block viral entry. Conversely, compounds that mimic ACE2 could competitively bind to the RBD, preventing the virus from attaching to host cells. Regulatory regions 437–508, and 681–686 were identified as critical for regulating viral entry. The RBD region (437–508) directly binds ACE2, while regulatory region 681–686 may influence the spike conformational state [24]. Limited proteolysis display sites (680–685 and 681–686) were identified as activation hotspots for the spike protein, essential for its conformational changes during membrane fusion [25]. The limited proteolysis display sites, essential for spike activation and viral membrane fusion, could be targeted using inhibitors that stabilize the spike protein in an inactive state. Small molecules that bind to these cleavage sites may interfere with protease accessibility, halting viral entry into host cells. This disordered nature of the region (437–508 and 681–686) allows it to act as an allosteric modulator, making it a promising candidate for small molecules designed to inhibit the conformational changes required for viral invasion.

Glycosylation display sites (16–21, 73–78, 148–153, 1157–1162, 1172–1177, and 1194–1198) are crucial for immune evasion, as they shield immunogenic epitopes from host immune detection. These sites are enriched in glycine, serine, and proline, which promote disorder, enabling the spike protein to adopt flexible structures that enhance immune evasion [26]. The glycosylation sites make them accessible for glycan addition, which impairs antibody recognition and contributes to the ability of the virus to evade immune responses. Targeting these regions could destabilize their shielding effects, exposing hidden immunogenic epitopes, facilitating the generation of neutralizing antibodies, and enhancing immune responses against the virus. This can improve the immune system’s ability to neutralize the virus. The intrinsic disorder observed in the SARS-CoV-2 spike protein offers unique opportunities for small-molecule drug discovery, particularly by targeting its disordered-to-ordered transitions and functionally relevant regions.

### 2.8. ORF3a Protein

The ORF3a of SARS-CoV-2 contains a significant portion of IDRs, which contribute to its multifunctional roles in viral pathogenesis and host-cell interactions. A proportion of 26.91% of its 275 amino acids consists of IDRs, which are particularly notable for its involvement in immune modulation, subcellular localization, and viral replication. These IDRs play pivotal roles in determining the protein’s functionality, as they enable dynamic interactions with the host cell machinery, influence subcellular trafficking, and modulate immune responses. The N-terminal domain (1–41) consistently exhibited high disorder propensity according to the VSL2 predictor, suggesting that this region coincides with the TRAF-binding motif (residues 36–40), which is implicated in protein–protein interactions. VSL2 predicted disorder at residues 1, 18–23, 169–183, and 221–225, which significantly overlap with the key experimental disorder regions (1–41). The ADOPT model recorded the highest Z-scores in the N-terminus (18–19, 160–181, 184–189, 195, 197, and 221), indicating ordered structures, whereas the lowest Z-score (below 3) was recorded in the C-terminal domain (235–275), indicating regions of high disorder, which does not align with experimental regions (1–41). VLXT also predicted relatively broad disordered regions, including 1–2, 127–130, 167–180, 221–222, and 263–275 (Figure 5). flDPnn identified disorder at 272 that did not align well with the experimentally verified region. Overall, VSL2 demonstrated the best alignment with experimental data by capturing a broader disordered region encompassing the validated segment. However, ADOPT, flDPnn, and VLXT failed to comprehensively capture the major experimentally validated disorder regions.

Research highlights the importance of the N-terminal region (1–41) in determining the sub-localization of ORF3a. Fluorescence microscopy experiments demonstrated that this IDR is vital for intracellular targeting and the retention of ORF3a at the plasma membrane or/and intracellular membranes [27]. It facilitates proper anchoring of ORF3a in the cytoplasm and plasma membrane, which is essential for viral assembly, immune evasion, and pathogenesis. These interactions could influence viral assembly, facilitate immune evasion by altering host cell signaling, or influence the inflammatory responses associated with SARS-CoV-2 infection. This IDR does not appear to interfere with the protein’s ion channel activity, but plays a vital role in intracellular trafficking, membrane association, and features common among viral proteins for their pathogenic efficiency. These findings suggest that targeting N-terminal IDR with small molecules can disrupt the protein’s ability to localize correctly, thereby attenuating its pathogenic functions and potentially reducing viral replication. This region’s evolutionary conservation across coronavirus further strengthens its role in viral pathogenesis and highlights its potential as a broad-spectrum antiviral target.

Another key IDR of ORF3a lies within residues 36–40, which are essential for binding to TRAF proteins (TRAF2/3/6) and play key roles in immune signaling by activating NF-κB and the NLRP3 inflammasome pathways involved in inflammation and apoptosis. Mutations that alter conserved residues in TRAF-binding motifs can reduce NF-κB activation and IL1β secretion, confirming the importance of this region in immune modulation [28]. The TRAF-binding motif facilitates the activation of the NLRP3 inflammasome, contributing to hyper-inflammation, a hallmark of severe COVID-19 infection. This IDR enables ORF3a to mediate key protein–protein interactions that regulate critical aspects of host immunity. Small molecules that disrupt the interaction of ORF3a with TRAF proteins could inhibit the activation of NF-κB pathway and prevent NLRP3 inflammasome assembly, thus mitigating the hyper-inflammatory response. This strategy allows for the reduction of cytokine secretion and alleviating severe immune reactions seen in COVID-19, such as cytokine storms and tissue damage.

### 2.9. Nucleocapsid (N) Protein

The N-protein comprises 419 amino acids, with 51% of its sequence being disordered. This disorder is primarily concentrated within the N-terminal domain (NTD), central linker/spacer region, and C-terminal domain (CTD). These disordered regions are implicated in RNA binding, genome packaging, liquid–liquid separation (LLPS), and protein–protein interaction, making them critical for viral replication and host interaction. Distinct disorder patterns were predicted across the N-terminal domain (1–68), central linker region (181–248), and C-terminus (370–419) by all three models (ADOPT, VSL2, and flDPnn), suggesting significant flexibility in these regions. A comparative analysis of disorder predictions from flDPnn, VSL2, VLXT, and ADOPT against experimentally validated disordered regions in the N-protein revealed significant differences in accuracy and coverage among the models. The experimental data identified three major disordered regions: 1–68, 171–248, and 362–419, which were used as benchmarks to assess model predictions. Among the predictive models, ADOPT partially covered the experimentally validated 1–68 region by predicting disorder from 1–47. It also partially overlapped with the 171–248 experimental region, though its predictions were fragmented (98, 172–216, 231, 234–268, and 270). Additionally, ADOPT aligned well with the 362–419 region, predicting disorder from 273–286, 365–419, and partially covered experimentally validated residues. VSL2 showed excellent predictive capability, identifying longer disordered segments, such as 1–84 in the N-terminal region and scattered residues (93–102, 142–143, 145–149, 163–289) in other regions. It performed exceptionally well in the 362–419 region, capturing 355–387, in agreement with the experimental findings. Notably, ADOPT and VSL2 consistently showed higher disorder scores compared with the other models, particularly in the NTD and CTD (Figure 6). flDPnn was the most conservative predictor, with minimal disorder assignments. It identified short disorder segments (1, 7, 11–12, 14, 16–19, 21–22, 24–25, 32, 35–57, and 59–75) in the N-terminal region and scattered residues (172–211, 213, 235, 237–267, 269, 275–277, 279–289, 368, 371, 373–383, and 397) across other regions. While flDPnn captured small disorder fragments, it lacked broader coverage of experimentally validated regions. VLXT correctly predicted disorder within 1–43 for the N-terminal region, and for the 171–248 region, it identified disorder in 73–102, 145–149, 152–160, 170, 172–215, and 325–271, with some overlap, but also significant fragmentation. In the 362–419 region, it captured 365–387, aligned with the experimental data. Overall, VSL2 demonstrated the best alignment with experimentally validated disorder regions, especially in the N-terminal and C-terminal regions, but still exhibited fragmentation in middle regions. ADOPT and VLXT provided a reasonable approximation of disorder, while flDPnn was more conservative in its predictions, often underestimating disorder. These findings underscore the value of using multiple prediction tools to gain a comprehensive understanding of the N-protein’s intrinsic disorder landscape and its potential implications for viral pathogenesis, including RNA binding, protein–protein interactions, and phase separation.

The NMR spectroscopy and fluorescence resonance energy transfer (FRET) experiments demonstrated that the NTD region (1–68) of N-protein plays a pivotal role in RNA binding and genome encapsidation. The NTD tail significantly influences the binding affinity of the N-protein for viral RNA, particularly poly(U) RNA [29]. Mutations within this region, such as proline to leucine substitution at position 13 and the deletion of residues (31–33) in Omicron variant, reduced the N-protein RNA binding affinity, highlighting the functional importance of these IDRs in viral replication [30]. This study observed a shift toward lower transfer efficiencies upon RNA binding, indicating that the IDR undergoes conformational changes upon RNA interaction. Small molecules targeting this IDR could disrupt the interaction between the N-protein and viral RNA, offering a potential therapeutic strategy to inhibit viral replication without the risk of rapid resistance development, as these regions are less prone to mutation compared with ordered regions.

The central linker region (181–248) plays a key role in liquid–liquid separation (LLPS), a process essential for the formation of ribonucleoprotein complexes during viral replication [31]. This phase separation allows N-protein to form dynamic liquid-like droplets that concentrate viral RNA and proteins, facilitating efficient genome packaging [32]. Disrupting this LLPS could interfere with virus replication and packaging. Small drug molecules may interfere with N-protein’s ability to undergo LLPS, potentially attenuating viral replication and providing a new avenue for antiviral therapy. The N-protein, as has molecular recognition site 186–190 vital for RNA binding moreover region 174–247, has been demonstrated to exhibit molecular condensation scaffolding activity [33].

The CTD region (370–419) plays an important role in N-protein oligomerization and the formation of biomolecular condensates. These condensates are membrane-less compartments that concentrate specific proteins and RNA molecules, critical for viral replication. Electron microscopy studies have demonstrated that this region is essential for forming spherical liquid droplets characteristic of biomolecular condensates. Deleting this region significantly reduced droplet formation, suggesting that its phase separation properties are essential for viral genome packaging and assembly. Small molecules targeting this region could disrupt the formation of these condensates, preventing viral replication by interfering with protein–protein and protein–RNA interactions. The high mutation rates observed in the disordered regions of the N-protein, particularly in the NTD, linker regions, and CTD, suggest that these regions contribute to immune evasion. Moreover, the IDRs could serve as epitopes for vaccine development. Inducing an immune response against these flexible regions may provide protective immunity, improving the efficacy of vaccines and helping the immune system to recognize the virus more effectively. These findings underscore the therapeutic potential of targeting IDRs in viral proteins, offering a novel strategy for antiviral drug development.

### 2.10. Statistical Analysis

The chi-squared test assessed the relationship between the experimental data and the predictions from four different disorder predictors (flDPnn, VLXT, VSL2, and ADOPT), as shown in Appendix A. The VSL2 model demonstrated the strongest performance, with an extremely low p-value and high chi-squared statistic (χ^2^ = 46.820, *p* < 0.0001). The ADOPT model also demonstrated a significant association (χ^2^ = 13.088, *p* = 0.0003), indicating a notable influence on the outcome. The significant *p*-value for flDPnn (*p* = 0.0347) suggests that there is a non-random association; while statistically significant, the relatively lower chi-squared value (4.458) indicates a moderate degree of association. The VLXT showed a stronger association (χ^2^ = 7.755, *p* = 0.0054), indicating a more pronounced relationship with the outcome. VSL2 was the best and most accurate predictor, followed by ADOPT, whereas the VLXT and flDPnn models were moderate predictors for our dataset.

### 2.11. Comparative Performance of Disorder Prediction Models

The predictive performance of the four disorder prediction models (ADOPT, VLXT, VSL2, and flDPnn) varied significantly based on different evaluation metrics, highlighting their strengths and limitations in identifying IDRs. For PONDR-based models, VSL2 outperformed VLXT, achieving a higher accuracy (82.3 ± 1.1%) and AUC (0.905) compared with VLXT’s accuracy of 69.0 ± 0.9% and AUC of 0.757. The chi-squared test (χ^2^ = 46.820, *p* < 0.0001) further confirmed VSL2’s highly significant association with the experimental data, reinforcing its reliability as the most accurate predictor in this study. ADOPT exhibited the highest predictive accuracy, with an AUC of 0.964 and a Matthews correlation coefficient (MCC) of 0.799, suggesting superior discriminatory power in distinguishing disordered and structured regions. ADOPT, despite having the highest AUC and a strong MCC, showed a lower chi-squared value (χ^2^ = 13.088, *p* = 0.0003) compared with VSL2. This suggests that, while ADOPT was highly sensitive in identifying disorder, its overall correlation with experimentally validated regions was slightly lower than that of VSL2. VLXT exhibited moderate performance, with an accuracy of 69.0 ± 0.9% and an AUC of 0.757. The chi-squared test (χ^2^ = 7.755, *p* = 0.0054) indicated a significant association with experimental data. In contrast, flDPnn, while still performing well (AUC = 0.814), exhibited a lower MCC of 0.370, indicating higher rates of false positives and false negatives compared with ADOPT, suggesting less robust predictive power. The chi-squared test (χ^2^ = 4.458, *p* = 0.0347) confirmed a statistically significant, but weaker, relationship with experimental disorder data compared with the other models. These results suggest that VSL2 is more reliable for disorder prediction than VLXT, as it better balances sensitivity and specificity.

VSL2 emerged as the most accurate predictor, exhibiting the strongest statistical association with experimentally validated IDRs. ADOPT, while having the highest AUC, showed slightly lower agreement with experimental results. VLXT and flDPnn provided moderate predictive capabilities, with flDPnn showing the weakest association. These findings emphasize the variability in predictive power among different models, highlighting the careful model selection depending on the dataset characteristics and the specific disorder prediction requirements. Collectively, VSL2 provided the most robust predictions, followed by ADOPT, while flDPnn and VLXT showed moderate predictive performance, with potential trade-offs in sensitivity and specificity.

### 2.12. Targeting IDRs in Drug Design

The feasibility of targeting intrinsically disordered regions (IDR) in drug discovery is evident, given their central role in protein–protein interactions (PPIs) and disease regulation. Unlike structured proteins, unstructured proteins, such as IDRs, provide dynamic binding sites that can be modulated by small-molecule inhibitors, peptide-based therapeutics, and targeted degradation strategies [34,35]. Small-molecule inhibitors, such as AJM589, can prevent oncogenic transcription by disrupting the c-Myc/Max interaction [36]. Similarly, MI-219 and Nutlin disrupt the p53–MDM2 interaction by binding to the MDM2 pocket [37]. MSI-1436, acting as an allosteric modulator, can induce conformational changes in C-PTIB, effectively inhibiting PTP1B function in breast cancer [38]. The PPI disruptor NSC635437 interferes with the EWS-FLI1 fusion protein, blocking RNA helicase binding and reducing tumorigenesis in Ewing’s sarcoma [39]. Peptide mimics, including those targeting p27-KID, competitively inhibit interactions crucial for cell cycle regulation, while an AF4-derived peptide prevents AF9 from binding properly, disrupting leukemia progression [40,41]. Drug repurposing efforts have identified trifluoperazine and fluphenazine as effective inhibitors of NUPR1, a key stress-response protein in pancreatic cancer [42]. Additionally, rational drug design has yielded ZZW-115, a compound specifically designed to target the hydrophobic IDR regions of NUPR1, offering a novel therapeutic approach for multiple cancers [36]. These advancements demonstrate the growing potential of IDR-targeted therapies in precision medicine and drug development.

## 3. Materials and Methods

The methodology employed in this research consisted of the selection of specific SARS-CoV-2 proteins, the utilization of bioinformatics tools to predict IDRs within these proteins, the scoring and ranking of disordered regions, and analysis as shown in Figure 7.

### 3.1. SARS-CoV-2 Protein Selection

Experimentally validated intrinsic disordered regions from the literature were utilized to assess the prediction of intrinsic disorder across a selection of SARS-CoV-2 proteins (Table 1). Using structural and functional data, such as cryo-EM and mass spectrometry evidence, enhanced the design of small-molecule inhibitors targeting IDRs. The proteins were chosen based on their roles in viral replication and immune modulation, and high disorder content in key regions depicted by knowledge-based literature search, which could be potential targets for therapeutic interventions. The sequences were retrieved from UniProt “https://www.uniprot.org/ (accessed on 27 August 2024)”, as shown in Appendix A. Disorder regions in these proteins were predicted using several disorder predictors.

### 3.2. Disorder Prediction Tools

Deep learning disorder prediction models, such as ADOPT, VLXT, VSL2, and flDPnn, were employed to analyze the intrinsic disorder propensity of selected SARS-CoV-2 proteins (Table 2). These models have been evaluated for accuracy within the critical assessment of protein structure prediction (CASP) [43]. These tools include both stand-alone predictors and meta-predictors, known for their high accuracy in predicting disordered regions. Meta-predictor was trained using the prediction outcomes from a group of predictors. Because the many predictors have information derived from various sequence properties, prediction models, and training sets, this method frequently leads to an obvious increase in prediction accuracy. Meta-predictors, VLXT, and VSL2 belong to the PONDR family and utilize a combination of neural networks to predict short and long disordered regions. VLXT utilizes a combination of three feedforward neural networks: VL1, XN, and XC. VL1 was trained on long disordered regions, while XN and XC were trained to identify short disordered regions at the N- and C-termini, respectively [44,45]. VSL2 used the linear support vector machine (SVM) to integrate predictions from individual predictors trained on datasets of different sequence length [46]. A length of 30 residues divided short and long IDRs with separate training sets for each class. VLXT is known for its high sensitivity to local sequences features, while VSL2 offers high accuracy for both short and long disordered regions.

Attention-based DisOrder PredicTor (ADOPT) is a deep learning model composed of a self-supervised encoder and a supervised disorder predictor [47]. The encoder extracts dense residue-level representations using Facebook’s evolutionary scale modeling (ESM) library, which is further processed by a supervised disorder predictor trained on a balanced dataset of ordered and disordered regions derived from NMR chemical shifts. ADOPT was trained on two datasets (CheZoD 1325 and CheZoD 117) to predict disorder with high accuracy, even for small protein sequences. It offers fast and accurate predictions compared with other existing methods. Another model, flDPnn, based on the neural network, was trained on 745 experimentally annotated proteins from the DisProt database (version 7.0) [48]. The dataset was split into training (445 proteins), validated (100 proteins), and test sets (200 proteins) to reduce overfitting and ensure accurate disorder predictions. It also participated in the CAID experiment, which evaluated 32 disorder predictors on a blind test of 646 proteins.

### 3.3. Disorder Scoring and Data Analysis

The disorder propensity of each residue in the selected proteins was predicted using the aforementioned tools. The output of each tool provides disorder scores, with higher scores indicating a greater likelihood of the region being intrinsically disordered. For VLXT, VSL2, and flDPnn, scores > 0.5 were considered indicative of disordered regions. Z-scores were calculated in ADOPT to quantify the deviation of the predicted disorder scores from the expected distribution for each residue. The Z-score is defined as the number of standard deviations that a given score deviates from the mean disorder score of a protein. Regions with a Z-score below 3 were considered to be fully disordered. Regions with Z-scores between 3 and 8 were classified as partially disordered. These regions may have some degree of structural order, but are still expected to be relatively flexible. Regions with Z-scores between 8 and 11 are considered to be flexible. These regions might have some degree of secondary structure, but are still expected to be dynamic and undergo conformational changes. Regions with Z-scores ≥ 11 were predicted to be structured (Table 3). These regions were expected to have a well-defined 3D structure, such as alpha-helices or beta-sheets.

### 3.4. Statistical Analysis

The statistical analysis was conducted using Python 3.6, employing the SciPy statistics library. A Chi-squared test was conducted to evaluate the relationship between experimentally validated and predicted disorder data from the ADOPT, VLXT, VSL2, and flDPnn models. This statistical test was employed to determine whether there was a significant association between the observed frequencies of disorder classifications and the expected frequencies under the assumption of independence. By comparing the distribution of predicted and experimentally validated data, the Chi-squared test allowed us to evaluate the degree of agreement between computational predictions and experimental data, thereby providing a quantitative measure of the models’ predictive reliability.

## 4. Conclusions

The analysis of intrinsically disordered regions (IDRs) in various SARS-CoV-2 proteins highlights their essential roles in viral replication, host interactions, and immune modulation. Due to their structural plasticity, IDRs enable dynamic protein conformations that facilitate crucial interactions with viral RNA, host cell membranes, and immune pathways. In the SARS-CoV-2 replicase polyprotein 1ab, multiple IDRs, including the flexible linker/spacer in NSP1, the Cu(II)-binding domain in NSP1, the RNA-binding domain in the polymerase, and the lipid-binding region in NSP6, are integral to viral replication and protein synthesis. Their structural flexibility and dynamic interactions with host factors make them attractive targets for small-molecule drug discovery. The disordered regions in the S protein, particularly in the receptor-binding domain (RBD), contribute to conformational flexibility, enhancing ACE2 receptor binding and viral entry, and may also play a role in immune evasion, contributing to variant emergence. IDRs in NSP11 undergo disorder-to-order transitions in lipid environments, facilitating viral replication through interactions with cellular membranes. These regions are critical to the assembly and stability of the viral replication machinery, making them a promising target for small-molecule inhibitors that could disrupt NSP11’s lipid-binding function and impair replication. The IDR in ORF3a is vital for protein localization and interactions with host proteins, influencing subcellular distribution and immune modulation. Its disordered segments also activate the NLRP3 inflammasome and NF-κB pathway, key players in the immune response and inflammation. Targeting these IDRs could help to inhibit SARS-CoV-2 replication, reduce immune hyperactivity, and improve patient outcomes. Disordered regions in NSP1 and NSP6 are also essential for viral translation and immune modulation through interactions with the ribosome and host immune factors, while disordered segments of NSP6 influence lipid binding and viral replication. These findings demonstrate the versatility and adaptive functions of IDRs in SARS-CoV-2 and emphasize their potential as broad-spectrum antiviral targets. Disrupting these dynamic IDRs could interfere with crucial viral processes, such as replication, protein synthesis, immune evasion, and host cell interactions. Given their evolutionary conservation across coronaviruses, these IDRs hold broad-spectrum antiviral potential, providing a promising therapeutic strategy to attenuate SARS-CoV-2 infections and possibly other related viral diseases.

## Figures and Tables

**Figure 1 ijms-26-03411-f001:**
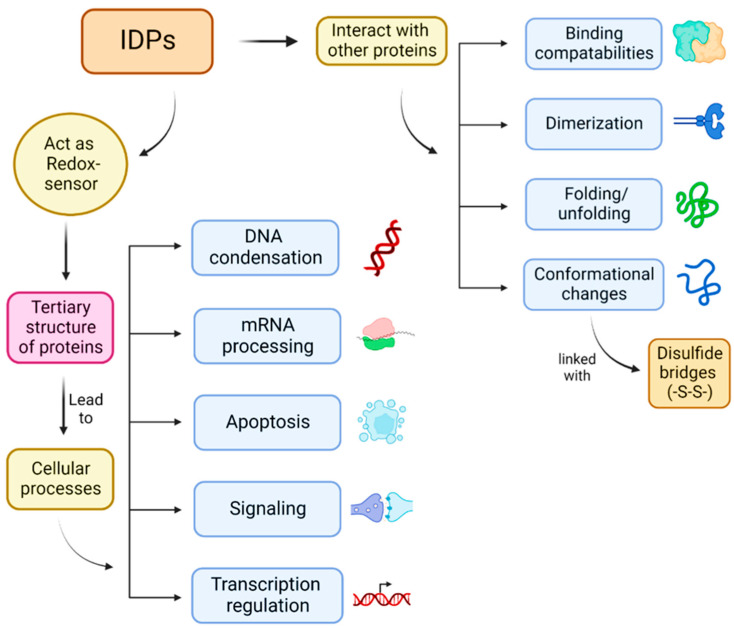
Intrinsically disordered proteins (IDPs): Significant role in several cellular processes.

**Figure 2 ijms-26-03411-f002:**
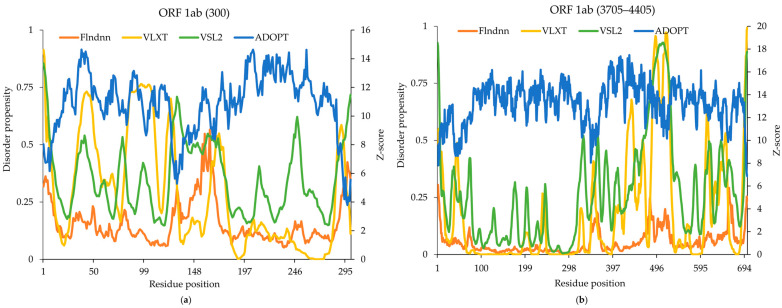
Predicted intrinsic disorder in SARS-CoV-2 replicase polyprotein 1ab. (**a**) Disorder propensity and Z-scores of the SARS-CoV-2 replicase polyprotein 1ab region (1–300); and (**b**) region (3501–4200) predicted by ADOPT, VLXT, VSL2, and flDPnn. Residue positions shown in the title were renumbered starting from 1 for the analysis.

**Figure 3 ijms-26-03411-f003:**
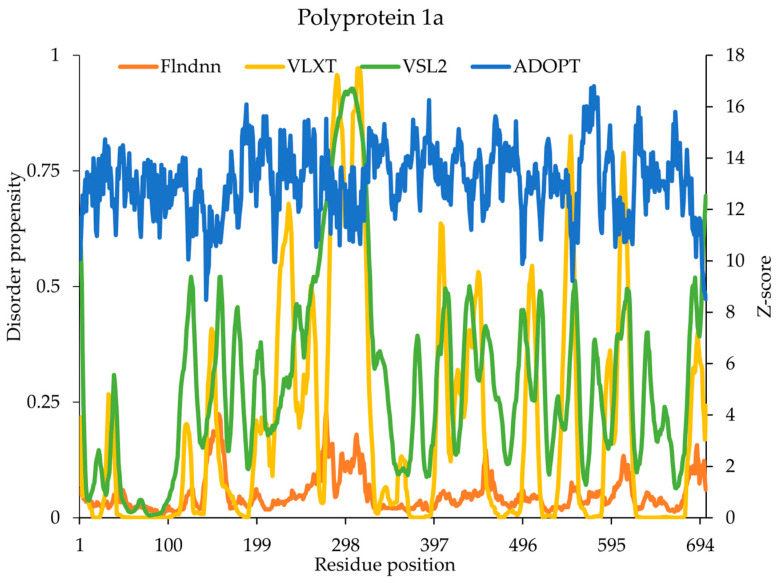
Predicted intrinsic disorder in SARS-CoV-2 replicase polyprotein 1a. Disorder propensity and Z-scores of the SARS-CoV-2 replicase polyprotein 1a predicted by ADOPT, VLXT, VSL2, and flDPnn. Residue positions shown in the title, which were renumbered starting from 1 for the analysis.

**Figure 4 ijms-26-03411-f004:**
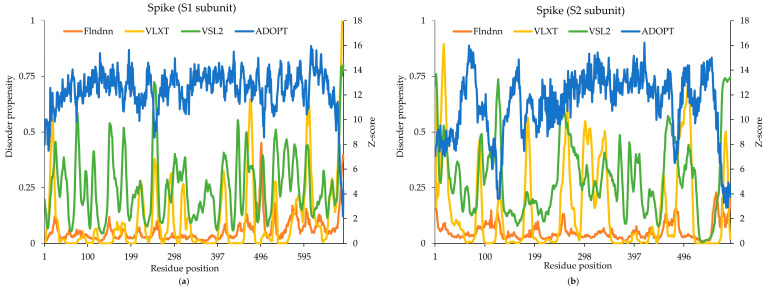
Predicted intrinsic disorder in SARS-CoV-2 (**a**) S1 subunit and (**b**) S2 subunit of spike glycoprotein. Disorder propensity and Z-scores of the SARS-CoV-2 spike were predicted by the ADOPT, VLXT, VSL2, and flDPnn models.

**Figure 5 ijms-26-03411-f005:**
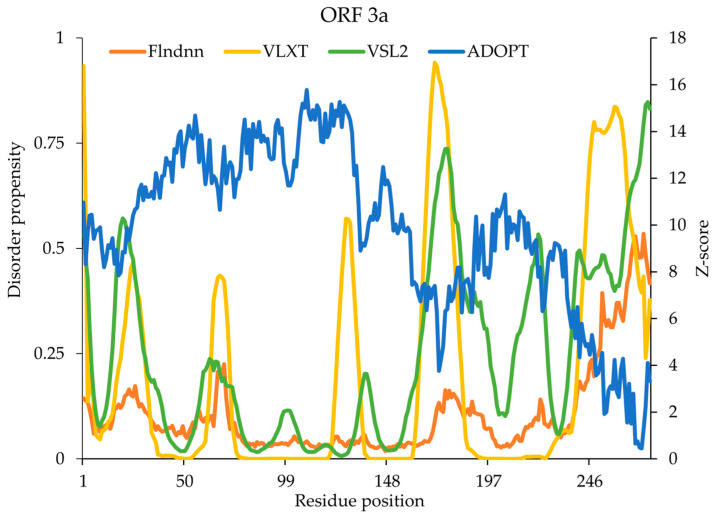
Predicted intrinsic disorder in the SARS-CoV-2 ORF3a protein. Disorder propensity and Z-scores of the spike protein predicted by ADOPT, flDPnn, VLXT, and VSL2.

**Figure 6 ijms-26-03411-f006:**
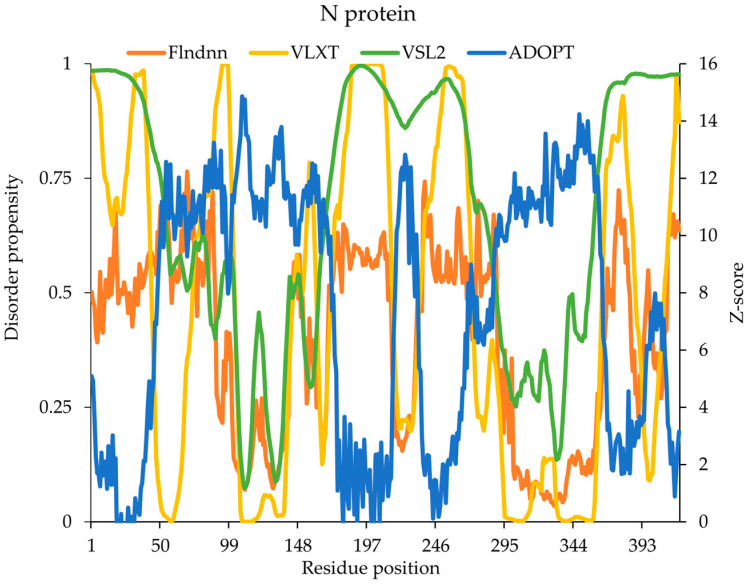
Predicted intrinsic disorder in SARS-CoV-2 N-protein. Disorder propensity and Z-scores of the N-protein predicted by ADOPT, VLXT, VSL2, and flDPnn.

**Figure 7 ijms-26-03411-f007:**
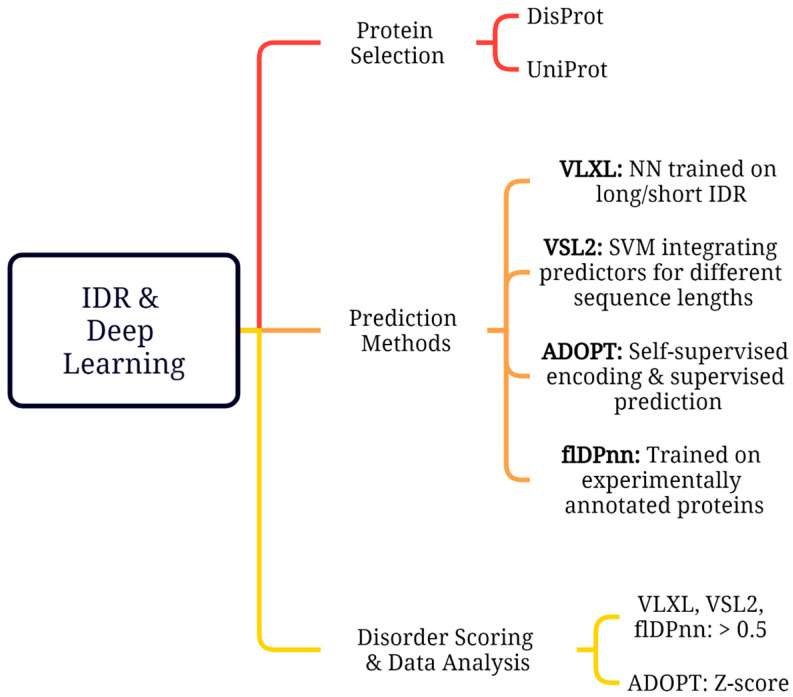
Schematic diagram of deep-learning-based approaches used for predicting IDRs.

**Table 1 ijms-26-03411-t001:** UniProt IDs for the prediction of IDRs across a selection of SARS-CoV-2 proteins.

UniProt	SARS-CoV-2 Proteins	Length	Disorder Content
P0DTC1	Replicase polyprotein 1a	4405	0.27%
P0DTC2	Spike glycoprotein (S)	1273	27.10%
P0DTC3	ORF3a protein	275	26.91%
P0DTC9	Nucleocapsid (N) protein	419	51.07%
P0DTD1	Replicase polyprotein 1ab	7096	4.61%

**Table 2 ijms-26-03411-t002:** Deep-learning-based disorder prediction tools such as ADOPT, VLXT, VSL2, and flDPnn used in this study.

Model	URL	Dataset	Algorithm	Properties	AUC	Evaluation Metrics	References
PONDR^®^VLXT	http://www.pondr.com/ (accessed on 25 September 2024)	Data consist of 8 and 7 long disordered regions from X-ray crystallography and NMR, respectively.	Three feedforward neural networks: VL1, XN, and XC	Less sensitive	0.757	ACC = 69.0 ± 0.9	[44,45]
PONDR^®^VSL2	http://www.pondr.com/ (accessed on 25 September 2024)	Sequences of varying lengths of X-ray crystallographic data (short and long length IDRs)	Combination of neural networks and SVM	81% accuracy reported	0.905	ACC = 82.3 ± 1.1	[46]
ADOPT	https://github.com/PeptoneLtd/ADOPT (accessed on 15 October 2024)	Datasets (CheZoD 1325 and CheZoD 117)	Deep bidirectional transformer (ESM library) and supervised predictor	Fast, accurate, and highly sensitive	0.964	MCC = 0.799	[47]
flDPnn	http://biomine.cs.vcu.edu/servers/flDPnn/ (accessed on 7 November 2024)	Annotated 745 proteins from the DisProt 7.0 database	Deep learning model	Fast	0.814	MCC = 0.370	[48]

Where AUC is the area under the ROC curve, ACC is accuracy and MCC is the Matthews correlation coefficient.

**Table 3 ijms-26-03411-t003:** Classification of protein regions into different disorder levels based on their Z-scores by ADOPT.

Disorder Level	Z-Score Range	Description
Fully Disordered	Z-score < 3	High degree of disorder, lacking a stable 3D structure
Partially Disordered	3 ≤ 8	Moderate disorder, some degree of structure but still flexible
Flexible	8 ≤ 11	Some degree of secondary structure but still dynamic
Structured	Z-score ≥ 11	Well-defined 3D structure, such as alpha-helices or beta-sheets

## Data Availability

The python code used in this study is available at “https://github.com/SI319” (accessed on 19 March 2025).

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
