# Peer review of "Deep Learning-Based Comparative Prediction and Functional Analysis of Intrinsically Disordered Regions in SARS-CoV-2"

_ijms, 2025, doi:10.3390/ijms26073411_

Round 1
Reviewer 1 Report
Comments and Suggestions for Authors
The manuscript presents a systematic analysis of intrinsically disordered regions (IDRs) in SARS-CoV-2 proteins using four deep learning models and validates predictions against existing wet-lab experimental data. While the comparative evaluation of prediction tools is thorough, the study’s central focus remains unclear. The title emphasizes “antiviral drug development,” yet the work primarily catalogs predicted IDRs and their functional annotations from prior literature (e.g., roles in replication or immune evasion). Critical challenges in targeting IDRs for drug discovery—such as amorphous binding pockets, dynamic conformational states, or strategies to stabilize/disrupt disordered regions—are not experimentally or computationally explored. This creates a mismatch between the stated aim (drug development) and the actual findings (IDR prediction and functional mapping). While the multi-model comparison is methodologically sound, its direct relevance to therapeutic design is underdeveloped. To strengthen novelty and alignment with the title, the authors should either reframe the narrative to highlight methodological benchmarking (e.g., tool performance for viral IDRs) or expand the discussion to address concrete drug-targeting strategies for IDRs (e.g., molecular docking simulations, ligand-induced folding assays, or case studies on existing IDR-targeting compounds). Clarifying the core scientific question—for instance, whether the study aims to benchmark predictive tools for viral IDRs or to propose actionable strategies for targeting IDRs in drug discovery—is essential. Aligning the title with the actual focus (e.g., ‘Comparative Prediction and Functional Analysis of Intrinsically Disordered Regions in SARS-CoV-2 Proteins’) would better reflect the work’s scope and prevent overstatement of therapeutic implications.
Author Response
We sincerely thank and appreciate the reviewer’s detailed feedback. We acknowledge that our primary contribution is the systematic analysis of IDRs in SARS-CoV-2 proteins, with a strong emphasis on benchmarking deep learning models for IDR prediction. However, we also highlight the therapeutic relevance of IDRs by integrating existing knowledge on druggable IDRs, PPI disruptors, and peptide mimics in results and discussion section (2.11. Targeting IDRs in drug design). This expanded discussion provides a concrete link between predicted IDRs and real-world drug discovery challenges, reinforcing the study's relevance to antiviral therapeutic development.
While our study does not conduct molecular docking (MD) simulations or ligand-induced folding assays, mapping IDRs across SARS-CoV-2 proteins is an essential first step in identifying potential targets for small-molecule inhibitors or peptide therapeutics. By validating IDR predictions against experimental data and reviewing known IDR-targeting strategies (Supplementary table S3), our work bridges computational predictions with actionable therapeutic insights, aligning with the study's stated aim.
Reviewer 2 Report
Comments and Suggestions for Authors
I have been reviewing the manuscript: “Deep learning-based prediction of disordered protein regions 2 in SARS-COV-2: a broad-spectrum antiviral drug design” that described the application of deep learning methods to the prediction of Intrinsically Disordered Regions in SARS-CoV-2 proteome.
According to the authors, this article should report about the assessment from four different Deep Learning -based predictive tools of the experimentally determined IDR regions in selected SARS-CoV-2 proteins, providing also insights into the function of the IDRs in SARS-CoV-2.
The research is interesting, considering the emerging importance of AI-based predictors in Structural Biology especially in of IDR description.
Nonetheless, I consider the present Manuscript not eligible for publication, at least in the present form, for the following reasons:
1) The Authors apply four different predictors, but nothing is said about their different general abilities to correctly predict structural disorder. No critical comparison is made of the four predictors, the outcome of the results is reported in figures that are not always really helpful (see later comment). Only disorder predictions from different methods are reported but the general ability of different software to predict experimentally determined IDR is not critically compared nor commented. How the different algorithms correctly predict the experimentally determined IDR? What kind of general conclusion we can gather from the application of the four methods here presented? Sometimes we observe very different results obtained from different methods, is this behavior related to the different algorithms? Any sequence-related dependence in the goodness of the predictions?
In general a critical analysis of the performances and an assessment of the different applied methods would be advisable.
2) Figures are of extreme importance in this Manuscript as they display the disorder predictions of the different methods. It is difficult to read these figures as they put everything on the same scale and nothing is said about different sensitivity of the methods. I wonder if the different methods require a re-normalization.
To make my point clear, according to what reported in Material and Methods, “VLXL (that is in figures and sometimes in text reported as VLXT), VSL2, and flDPnn scores >0.5 were considered indicative of disordered regions” but in this case VLXL never reach 0.5 so it seems completely useless as disorder predictors, is it true or there is a problem in scaling different outcome together? Apart from N-protein also flDPnn exceeds 0.5 in very few cases with respect to VSL2 and ADOPT. Are the score properly normalized and then comparable? Moreover, the same numbering of the sequence should be used in both figures and text, which is not always true.
3) Several statements are made in the manuscript that are not always supported by proper publications nor the source of the data are clearly cited.
Author Response
Comment 1: The Authors apply four different predictors, but nothing is said about their different general abilities to correctly predict structural disorder. No critical comparison is made of the four predictors, the outcome of the results is reported in figures that are not always really helpful (see later comment). Only disorder predictions from different methods are reported but the general ability of different software to predict experimentally determined IDR is not critically compared nor commented. How the different algorithms correctly predict the experimentally determined IDR? What kind of general conclusion we can gather from the application of the four methods here presented? Sometimes we observe very different results obtained from different methods, is this behavior related to the different algorithms? Any sequence-related dependence in the goodness of the predictions?
In general, a critical analysis of the performances and an assessment of the different applied methods would be advisable.
Response 1: We sincerely apologize regarding the critical comparison and analysis of the four disorder predictors. We recognize the importance of assessing their individual abilities and providing a comprehensive evaluation of their performance against experimentally determined IDR. We appreciate the reviewer's insightful observations regarding the comparative performance of the disorder prediction models. We have now included a critical evaluation of the four predictors in our “Results and discussion section” highlighting which models show higher reliability for specific protein regions. We also performed Chi-squared test analysis which quantifies the agreement between predictions and experimental data, allowing us to assess the statistical significance of each predictor's accuracy. We acknowledge that differences in prediction arise due to the varying algorithmic approaches. We also addressed certain sequences tend to be over- or under-predicted by specific models. We highlighted specific cases where one predictor outperforms others, guiding future studies in selecting appropriate tools based on their intended applications. We believe these refinements significantly enhance the manuscript's clarity and depth.
Comment 2: Figures are of extreme importance in this Manuscript as they display the disorder predictions of the different methods. It is difficult to read these figures as they put everything on the same scale and nothing is said about different sensitivity of the methods. I wonder if the different methods require a re-normalization.
To make my point clear, according to what reported in Material and Methods, “VLXL (that is in figures and sometimes in text reported as VLXT), VSL2, and flDPnn scores >0.5 were considered indicative of disordered regions” but in this case VLXL never reach 0.5 so it seems completely useless as disorder predictors, is it true or there is a problem in scaling different outcome together? Apart from N-protein also flDPnn exceeds 0.5 in very few cases with respect to VSL2 and ADOPT. Are the score properly normalized and then comparable? Moreover, the same numbering of the sequence should be used in both figures and text, which is not always true.
Response 2: We appreciate the reviewer’s observation regarding the thresholding of disorder propensity scores and the apparent discrepancy in PONDR VLXT’s predictive performance. Upon further review, we identified that VLXT data was initially misinterpreted due to some missing values, leading to an underestimation of its predictive capability. After correcting this, we observed improved results for VLXT, confirming that it provides meaningful disorder predictions (Results and Discussion section). While applied normalized data for the Chi-squared test, the raw propensity scores were not initially scaled together. Furthermore, we agree that flDPnn also tends to provide lower disorder propensity scores compared to other models and only exceeds the 0.5 threshold in relatively few cases, except for the N-protein. This difference reflects variation in disorder prediction algorithms rather than an inherent weakness of flDPnn. Given that flDPnn was trained on different datasets and employs distinct machine learning techniques, its propensity score distribution differs from other models. We have included a supplementary table S1 to renumber IDRs segments more clearly.
Comment 3: Several statements are made in the manuscript that are not always supported by proper publications nor the source of the data are clearly cited.
Response 3: We thank the reviewer for their careful review and for pointing out the instances where our statements lacked sufficient support. We have addressed each of these issues in the revised manuscript.
Reviewer 3 Report
Comments and Suggestions for Authors
I thought this was an excellent paper, and I thoroughly enjoyed reading it. The research offers valuable insights into the prediction of disordered protein regions in SARS-CoV-2, which is relevant and timely. However, I believe that certain sections of the manuscript can be enhanced to explain and solidify the findings better:
Simplification of Charts: While the tables and figures are presented succinctly enough to clearly express results and methodology, some can be simplified for clearer understanding. As an example, creating a summary table that underlines the comparative prediction of disorder in different models for significant regions of proteins would enable readers to understand the main findings easily.
Statistical Analysis: A more detailed statistical analysis would serve to tighten the bond between the experimentally verified and predicted disordered areas in the manuscript. Describing p-values, confidence limits, or regression analysis would really serve to build much stronger supports for the prediction models used.
Wider Implications: The case for the feasibility of targeting IDRs in drug design is hopeful. It would be improved if the section was longer to talk about how such discoveries can be applied to develop real therapeutic strategies. Potential strategies such as the creation of small molecule inhibitors or peptides that mimic disordered segments can provide a more positive view of the implications of the research.
Methodological Details: It would be useful to give further description of deep learning models that have been used for disorder prediction. Including the comparison of performance metrics of each model such as accuracy, sensitivity, and specificity would give a more logical rationale for their selection and application in your study.
Experimental Validation: Relying on predictions from bioinformatics software may be supplemented with experimental validation to validate the findings of the study. If available, the inclusion of information regarding peptide synthesis and characterization of disorder using methods like spectroscopy or microscopy would validate the predictions. I understand that experimental validation may not yet be available, but any additional data would make the manuscript significantly better.
Author Response
We sincerely appreciate the reviewer's thoughtful and constructive feedback.
- We have created a summary table (Supplementary Table S1) that highlights the comparative prediction of disorder across different models for key protein regions. This addition enhances readability and allows for a clearer understanding of the main findings.
- We have performed detailed statistical analyses to assess the correlation between predicted intrinsically disordered regions (IDRs) and experimentally verified disordered regions (Supplementary Table S2, S3).
- We have extended the discussion on the feasibility of targeting IDRs in drug design, elaborating on the potential of small molecule inhibitors and peptide mimetics as therapeutic strategies in results and discussion section (2.11. Targeting IDRs in Drug Design). This expansion provides a more comprehensive perspective on how our findings can be translated into real-world drug development efforts.
- To provide a more logical rationale for model selection, we have included AUC scores for each deep learning model used in the study (Table 2).
- We fully acknowledge the importance of experimental validation in reinforcing our computational findings. While direct experimental characterization, such as peptide synthesis and disorder assessment via spectroscopy or microscopy, remains beyond the current scope due to computational costs and the large-scale nature of our dataset, we recognize its value for future studies.
We are grateful for the reviewer’s constructive feedback, which has significantly contributed to refining our manuscript.